# Identifying Causal-Effect Inference Failure with Uncertainty-Aware Models

**Andrew Jesson**[*]
Department of Computer Science
University of Oxford
Oxford, UK OX1 3QD
andrew.jesson@cs.ox.ac.uk

**Sören Mindermann**[*]
Department of Computer Science
University of Oxford
Oxford, UK OX1 3QD
soren.mindermann@cs.ox.ac.uk

**Uri Shalit**
Technion
Haifa, Israel 3200003
urishalit@technion.ac.il

**Yarin Gal**
Department of Computer Science
University of Oxford
Oxford, UK OX1 3QD
yarin.gal@cs.ox.ac.uk

## Abstract

Recommending the best course of action for an individual is a major application of individual-level causal effect estimation. This application is often needed in safety-critical domains such as healthcare, where estimating and communicating uncertainty to decision-makers is crucial. We introduce a practical approach for integrating uncertainty estimation into a class of state-of-the-art neural network methods used for individual-level causal estimates. We show that our methods enable us to deal gracefully with situations of "no-overlap", common in high-dimensional data, where standard applications of causal effect approaches fail. Further, our methods allow us to handle covariate shift, where the train and test distributions differ, common when systems are deployed in practice. We show that when such a covariate shift occurs, correctly modeling uncertainty can keep us from giving overconfident and potentially harmful recommendations. We demonstrate our methodology with a range of state-of-the-art models. Under both covariate shift and lack of overlap, our uncertainty-equipped methods can alert decision makers when predictions are not to be trusted while outperforming standard methods that use the propensity score to identify lack of overlap.

## 1   Introduction

Learning individual-level causal effects is concerned with learning how units of interest respond to interventions or treatments. These could be the medications prescribed to particular patients, training-programs to job seekers, or educational courses for students. Ideally, such causal effects would be estimated from randomized controlled trials, but in many cases, such trials are unethical or expensive: researchers cannot randomly prescribe smoking to assess health risks. Observational data offers an alternative, with typically larger sample sizes and lower costs, and more relevance to the target population. However, the price we pay for using observational data is lower certainty in our causal estimates, due to the possibility of unmeasured confounding, and the measured and unmeasured differences between the populations who were subject to different treatments.

Progress in learning individual-level causal effects is being accelerated by deep learning approaches to causal inference [27, 36, 3, 48]. Such neural networks can be used to learn causal effects from

---
[*]Equal contribution.

observational data, but current deep learning tools for causal inference cannot yet indicate when they are unfamiliar with a given data point. For example, a system may offer a patient a recommendation even though it may not have learned from data belonging to anyone with similar age or gender as the patient, or it may have never observed someone like this patient receive a specific treatment before. In the language of machine learning and causal inference, the first example corresponds to a *covariate shift*, and the second example corresponds to a violation of the *overlap assumption*, also known as positivity. When a system experiences either covariate shift or violations of overlap, the recommendation would be uninformed and could lead to undue stress, financial burden, false hope, or worse. In this paper, we explain and examine how covariate shift and violations of overlap are concerns for the real-world use of learning conditional average treatment effects (CATE) from observational data, why deep learning systems should indicate their lack of confidence when these phenomena are encountered, and develop a new and principled approach to incorporating uncertainty estimating into the design of systems for CATE inference.

First, we reformulate the lack of overlap at test time as an instance of covariate shift, allowing us to address both problems with one methodology. When an observation $\boldsymbol{x}$ lacks overlap, the model predicts the outcome $y$ for a treatment $t$ that has probability zero or near-zero under the training distribution. We extend the Causal-Effect Variational Autoencoder (CEVAE) [36] by introducing a method for out-of-distribution (OoD) training, negative sampling, to model uncertainty on OoD inputs. Negative sampling is effective and theoretically justified but usually intractable [18]. Our insight is that it becomes tractable for addressing non-overlap since the distribution of test-time inputs $(\boldsymbol{x}, t)$ is known: it equals the training distribution but with a different choice of treatment (for example, if at training we observe outcome $y$ for patient $\boldsymbol{x}$ only under treatment $t = 0$, then we know that the outcome for $(\boldsymbol{x}, t = 1)$ should be uncertain). This can be seen as a special case of transductive learning [57, Ch. 9]. For addressing covariate shift in the inputs $\boldsymbol{x}$, negative sampling remains intractable as the new covariate distribution is unknown; however, it has been shown in non-causal applications that Bayesian parameter uncertainty captures "epistemic" uncertainty which can indicate covariate shift [29]. We, therefore, propose to treat the decoder $p(y|\boldsymbol{x}, t)$ in CEVAE as a Bayesian neural network able to capture epistemic uncertainty.

In addition to casting lack of overlap as a distribution shift problem and proposing an OoD training methodology for the CEVAE model, we further extend the modeling of epistemic uncertainty to a range of state-of-the-art neural models including TARNet, CFRNet [47], and Dragonnet [49], developing a practical Bayesian counter-part to each. We demonstrate that, by excluding test points with high epistemic uncertainty at test time, we outperform baselines that use the propensity score $p(t = 1|\boldsymbol{x})$ to exclude points that violate overlap. This result holds across different state-of-the-art architectures on the causal inference benchmarks IHDP [23] and ACIC [11]. Leveraging uncertainty for exclusion ties it into causal inference practice where a large number of overlap-violating points must often be discarded or submitted for further scrutiny [43, 25, 6, 26, 20]. Finally, we introduce a new semi-synthetic benchmark dataset, CEMNIST, to explore the problem of non-overlap in high-dimensional settings.

## 2 Background

Classic machine learning is concerned with functions that map an input (e.g. an image) to an output (e.g. "is a person"). The specific function $f$ for a given task is typically chosen by an algorithm that minimizes a loss between the outputs $f(\boldsymbol{x}_i)$ and targets $y_i$ over a dataset $\{\boldsymbol{x}_i, y_i\}_{i=1}^{N}$ of input covariates and output targets. Causal effect estimation differs in that, for each input $\boldsymbol{x}_i$, there is a corresponding treatment $t_i \in \{0, 1\}$ and two potential outcomes $Y^1, Y^0$ – one for each choice of treatment [45]. In this work, we are interested in the Conditional Average Treatment Effect (CATE):

$$\text{CATE}(\boldsymbol{x}_i) = \mathbb{E}[Y^1 - Y^0 | \boldsymbol{X} = \boldsymbol{x}_i] \tag{1}$$

$$= \mu^1(\boldsymbol{x}_i) - \mu^0(\boldsymbol{x}_i), \tag{2}$$

where the expectation is needed both because the *individual treatment effect* $Y^1 - Y^0$ may be non-deterministic, and because it cannot in general be identified without further assumptions. Under the assumption of ignorability conditioned on $\boldsymbol{X}$ (or no-hidden confounding) which we make in this paper, we have that $\mathbb{E}[Y^a | \boldsymbol{X} = \boldsymbol{x}_i] = \mathbb{E}[y | \boldsymbol{X} = \boldsymbol{x}_i, t = a]$, thus opening the way to estimate CATE from observational data [26]. Specifically, we are motivated by cases where $\boldsymbol{X}$ is high-dimensional, for example, a patient's entire medical record, in which case we can think of the CATE as representing an individual-level causal effect. Though the specific meaning of a CATE measurement depends on

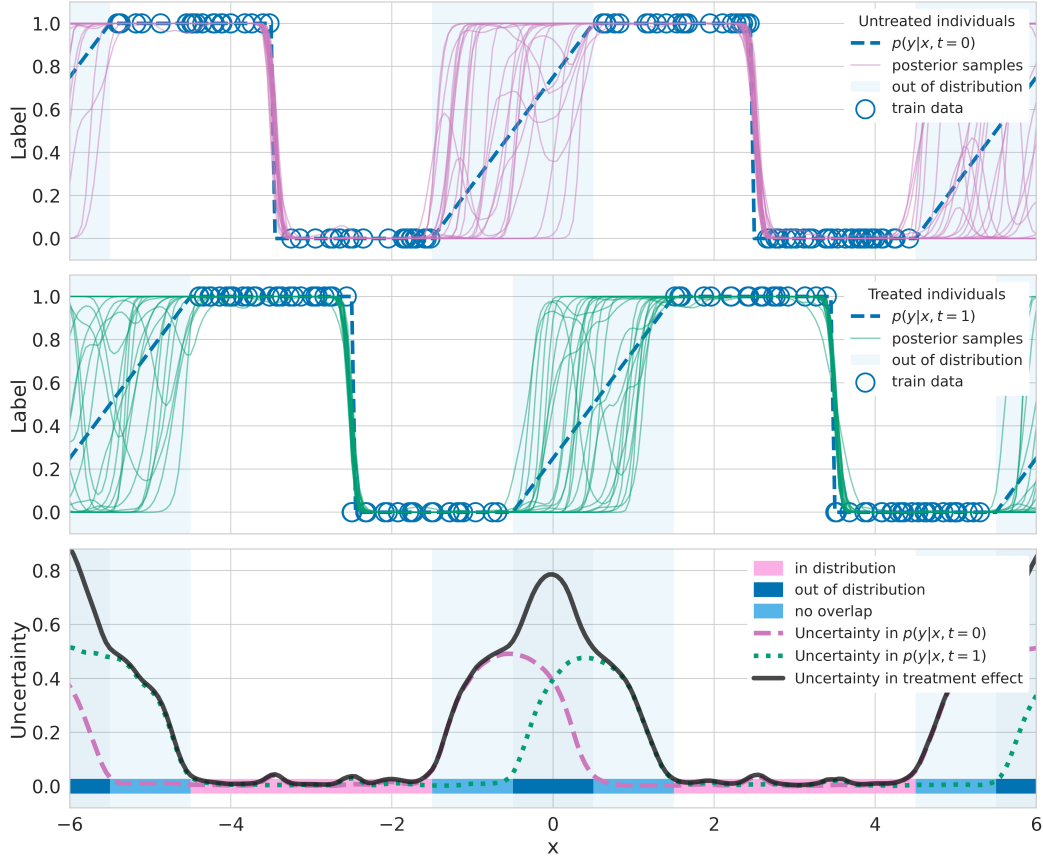

Figure 1: Explanation how epistemic uncertainty detects lack of data. **Top**: binary outcome $y$ (blue circle) given *no* treatment, and different functions $p(y = 1|\boldsymbol{x}, t = 0, \boldsymbol{\omega})$ (purple) predicting outcome probability (blue dashed line, ground truth). Functions disagree where data is scarce. **middle**: binary outcome $y$ given treatment, and functions $p(y = 1|\boldsymbol{x}, t = 1, \boldsymbol{\omega})$ (green) predicting outcome probability. **Bottom**: measures of uncertainty/disagreement between outcome predictions (dashed purple and dotted green lines) are high when data is lacking. CATE uncertainty (solid black line) is higher where at least one model lacks data (non-overlap, light blue) or where both lack data (out-of-distribution / covariate shift, dark blue).

context, in general, a positive value indicates that an individual with covariates $\boldsymbol{x}_i$ will have a positive response to treatment, a negative value indicates a negative response, and a value of zero indicates that the treatment will not affect such an individual.

The fundamental problem of learning to infer CATE from an observational dataset $\mathcal{D} = \{\boldsymbol{x}_i, y_i, t_i\}_{i=1}^{N}$ is that only the *factual* outcome $y_i = Y^{t_i}$ corresponding to the treatment $t_i$ can be observed. Because the *counterfactual* outcome $Y^{1-t_i}$ is never observed, it is difficult to learn a function for $\text{CATE}(\boldsymbol{x}_i)$ directly. Instead, a standard approach is often either to treat $t_i$ as an additional covariate [16] or focus on learning functions for $\mu^0(\boldsymbol{x}_i)$ and $\mu^1(\boldsymbol{x}_i)$ using the observed $y_i$ in $\mathcal{D}$ as targets [47, 36, 48].

## 2.1 Epistemic uncertainty and covariate shift

In probabilistic modelling, predictions may be assumed to come from a graphical model $p(y|\boldsymbol{x}, t, \boldsymbol{\omega})$ – a distribution over outputs (the likelihood) given a single set of parameters $\boldsymbol{\omega}$. Considering a binary label $y$ given, for example, $t = 0$, a neural network can be described as a function defining the likelihood $p(y = 1|\boldsymbol{x}, t = 0, \boldsymbol{\omega}_0)$, with parameters $\boldsymbol{\omega}_0$ defining the network weights. Different draws $\boldsymbol{\omega}_0$ from a distribution over parameters $p(\boldsymbol{\omega}_0|\mathcal{D})$ would then correspond to different neural networks, i.e. functions from $(\boldsymbol{x}, t = 0)$ to $y$ (e.g. the purple curves in Fig. 1 (top)).

For parametric models such as neural networks (NNs), we treat the weights as random variables, and, with a chosen prior distribution $p(\boldsymbol{\omega}_0)$, aim to infer the posterior distribution $p(\boldsymbol{\omega}_0|\mathcal{D})$. The purple curves in Figure 1 (top) are individual NN's $\mu^{\boldsymbol{\omega}_0}(\cdot)$ sampled from the posterior of such a Bayesian neural network (BNN). *Bayesian inference* can be performed by marginalizing the likelihood function $p(y|\mu^{\boldsymbol{\omega}_0}(\boldsymbol{x}))$ over the posterior $p(\boldsymbol{\omega}_0|\mathcal{D})$ in order to obtain the posterior predictive probability $p(y|\boldsymbol{x}, t = 0, \mathcal{D}) = \int p(y|\boldsymbol{x}, t = 0, \boldsymbol{\omega}_0)p(\boldsymbol{\omega}_0|\mathcal{D})d\boldsymbol{\omega}_0$. This marginalization is intractable for BNNs in practice, so variational inference is commonly used as a scalable approximate inference technique, for example, by sampling the weights from a Dropout approximate posterior $q(\boldsymbol{\omega}_0|\mathcal{D})$ [15].

Figure 1 (top) illustrates the effects of a BNN's parameter uncertainty in the range $\boldsymbol{x} \in [-1, 1]$ (shaded region). While all sampled functions $\mu^{\boldsymbol{\omega}_0}(\boldsymbol{x})$ with $\boldsymbol{\omega}_0 \sim p(\boldsymbol{\omega}_0|\mathcal{D}, t = 0)$ (shown in blue) agree with each other for inputs $\boldsymbol{x}$ in-distribution ($\boldsymbol{x} \in [-6, -1]$) these functions make disagreeing predictions for inputs $\boldsymbol{x} \in [-1, 1]$ because these lie out-of-distribution (OoD) with respect to the training distribution $p(\boldsymbol{x}|t = 0)$. This is an example of *covariate shift*.

To avoid overconfident erroneous extrapolations on such OoD examples, we would like to indicate that the prediction $\mu^{\boldsymbol{\omega}_0}(\boldsymbol{x})$ is uncertain. This *epistemic* uncertainty stems from a lack of data, not from measurement noise (also called *aleatoric* uncertainty). Epistemic uncertainty about the random variable (r.v.) $Y^0$ can be quantified in various ways. For classification tasks, a popular information-theoretic measure is the information gained about the r.v. $\boldsymbol{\omega}_0$ if the label $y = Y^0$ were observed for a new data point $\boldsymbol{x}$, given the training dataset $\mathcal{D}$ [24]. This is captured by the mutual information between $\boldsymbol{\omega}_0$ and $Y^0$, given by

$$\mathcal{I}[\boldsymbol{\omega}_0, Y^0|\mathcal{D}, \boldsymbol{x}] = \mathcal{H}[Y^0|\boldsymbol{x}, \mathcal{D}] \quad - \underset{q(\boldsymbol{\omega}_0|\mathcal{D})}{\mathbb{E}} \left[ \mathcal{H}[Y^0|\boldsymbol{x}, \boldsymbol{\omega}_0] \right], \tag{3}$$

where $\mathcal{H}[\bullet]$ is the entropy of a given r.v. For regression tasks, it is common to measure how the r.v. $\mu^{\boldsymbol{\omega}_0}(\boldsymbol{x})$ varies when marginalizing over $\boldsymbol{\omega}_0$: $\underset{q(\boldsymbol{\omega}_0|\mathcal{D})}{\mathrm{Var}} [\mu^{\boldsymbol{\omega}_0}(\boldsymbol{x})]$. We will later use this measure for epistemic uncertainty in CATE.

## 3 Non-overlap as a covariate shift problem

Standard causal inference tasks, under the assumption of ignorability conditioned on $\boldsymbol{X}$, usually deal with estimating both $\mu^0(\boldsymbol{x}) = \mathbb{E}[y|\boldsymbol{X} = \boldsymbol{x}, t = 0]$ and $\mu^1(\boldsymbol{x}) = \mathbb{E}[y|\boldsymbol{X} = \boldsymbol{x}, t = 1]$. Overlap is usually assumed as a means to address this problem. The overlap assumption (also known as *common support* or *positivity*) states that there exists $0 < \eta < 0.5$ such that the *propensity score* $p(t = 1|\boldsymbol{x})$ satisfies:

$$\eta < p(t = 1|\boldsymbol{x}) < 1 - \eta, \tag{4}$$

i.e., that for every $\boldsymbol{x} \sim p(\boldsymbol{x})$, we have a non-zero probability of observing its outcome under $t = 1$ as well as under $t = 0$. This version is sometimes called *strict overlap*, see [8] for discussion. When overlap does not hold for some $\boldsymbol{x}$, we might lack data to estimate either $\mu^0(\boldsymbol{x})$ or $\mu^1(\boldsymbol{x})$—this is the case in the grey shaded areas in Figure 1 (bottom).

Overlap is a central assumption in causal inference [43, 25]. Nonetheless, it is usually not satisfied for all units in a given observational dataset [43, 25, 6, 26, 20]. It is even harder to satisfy for high-dimensional data such as images and comprehensive demographic data [8] where neural networks are used in practice [17].

Since overlap must be assumed for most causal inference methods, an enormously popular practice is "trimming": removing the data points for which overlap is not satisfied before training [20, 13, 48, 30, 7]. In practice, points are trimmed when they have a propensity close to 0 or 1, as predicted by a trained propensity model $p^{\boldsymbol{\omega}_p}(t|\boldsymbol{x})$. The average treatment effect (ATE), is then calculated by over the remaining training points.

However, trimming has a different implication when estimating the CATE for each unit with covariates $\boldsymbol{x}_i$: it means that for some units a CATE estimate is not given. If we think of CATE as a tool for recommending treatment assignment, a trimmed unit receives no treatment recommendation. This reflects the uncertainty in estimating one of the potential outcomes for this unit, since treatment was rarely (if ever) given to similar units. In what follows, we will explore how trimming can be replaced

with more data-efficient rejection methods that are specifically focused on assessing the level of uncertainty in estimating the expected outcomes for $\boldsymbol{x}_i$ under both treatment options.

Our model of the CATE is:

$$\widehat{\mathrm{CATE}}^{\boldsymbol{\omega}_{0/1}}(\boldsymbol{x}) = \mu^{\boldsymbol{\omega}_1}(\boldsymbol{x}) - \mu^{\boldsymbol{\omega}_0}(\boldsymbol{x}). \tag{5}$$

In Figure 1, we illustrate that lack of overlap constitutes a covariate shift problem. When $p(t = 1|\boldsymbol{x}_{\mathrm{test}}) \approx 0$, we face a covariate shift for $\mu^{\boldsymbol{\omega}_1}(\cdot)$ (because by Bayes rule $p(\boldsymbol{x}_{\mathrm{test}}|t = 1) \approx 0$). When $p(t = 1|\boldsymbol{x}_{\mathrm{test}}) \approx 1$, we face a covariate shift for $\mu^{\boldsymbol{\omega}_0}(\cdot)$, and when $p(\boldsymbol{x}_{\mathrm{test}}) \approx 0$, we face a covariate shift for $\widehat{\mathrm{CATE}}^{\boldsymbol{\omega}_{0/1}}(\boldsymbol{x})$ ("out of distribution" in the Figure 1 (bottom)). With this understanding, we can deploy tools for epistemic uncertainty to address both covariate shift and non-overlap simultaneously.

### 3.1 Epistemic uncertainty in CATE

To the best of our knowledge, uncertainty in high-dimensional CATE (i.e. where each value of $\boldsymbol{x}$ is only expected to be observed at most once) has not been previously addressed. $\mathrm{CATE}(\boldsymbol{x})$ can be seen as the first moment of the random variable $Y^1 - Y^0$ given $\boldsymbol{X} = \boldsymbol{x}$. Here, we extend this notion and examine the *second* moment, the variance, which we can decompose into its aleatoric and epistemic parts by using the law of total variance:

$$\underset{p(\boldsymbol{\omega}_0, \boldsymbol{\omega}_1, Y^0, Y^1|\mathcal{D})}{\mathrm{Var}}[Y^1 - Y^0|\boldsymbol{x}] = \underset{p(\boldsymbol{\omega}_0, \boldsymbol{\omega}_1|\mathcal{D})}{\mathbb{E}}\left[\underset{Y_0, Y_1}{\mathrm{Var}}\left[Y^1 - Y^0 \mid \mu^{\boldsymbol{\omega}_1}(\boldsymbol{x}), \mu^{\boldsymbol{\omega}_0}(\boldsymbol{x})\right]\right] \\ + \underset{p(\boldsymbol{\omega}_0, \boldsymbol{\omega}_1|\mathcal{D})}{\mathrm{Var}}[\mu^{\boldsymbol{\omega}_1}(\boldsymbol{x}) - \mu^{\boldsymbol{\omega}_0}(\boldsymbol{x})]. \tag{6}$$

The second term on the r.h.s. is $\mathrm{Var}[\widehat{\mathrm{CATE}}^{\boldsymbol{\omega}_{0/1}}(\boldsymbol{x})]$. It measures the epistemic uncertainty in CATE since it only stems from the disagreement between predictions for different values of the parameters, not from noise in $Y^1, Y^0$. We will use this uncertainty in our methods and estimate it directly by sampling from the approximate posterior $q(\boldsymbol{\omega}_0, \boldsymbol{\omega}_1|\mathcal{D})$. The first term on the r.h.s. is the expected aleatoric uncertainty, which is disregarded in CATE estimation (but could be relevant elsewhere).

Referring back to Figure 1, when overlap is not satisfied for $\boldsymbol{x}$, $\mathrm{Var}[\widehat{\mathrm{CATE}}^{\boldsymbol{\omega}_{0/1}}(\boldsymbol{x})]$ is large because at least one of $\mathrm{Var}_{\boldsymbol{\omega}_0}[\mu^{\boldsymbol{\omega}_0}(\boldsymbol{x})]$ and $\mathrm{Var}_{\boldsymbol{\omega}_1}[\mu^{\boldsymbol{\omega}_1}(\boldsymbol{x})]$ is large. Similarly, under regular covariate shift ($p(\boldsymbol{x}) \approx 0$), both will be large.

### 3.2 Rejection policies with epistemic uncertainty versus propensity score

If there is insufficient knowledge about an individual, and a high cost associated with making errors, it may be preferable to withhold the treatment recommendation. It is therefore important to have an informed *rejection policy*. In our experiments, we reject, i.e. choose to make no treatment recommendation, when the epistemic uncertainty exceeds a certain threshold. In general, setting the threshold will be a domain-specific problem that depends on the cost of type I (incorrectly recommending treatment) and type II (incorrectly withholding treatment) errors. In the diagnostic setting, thresholds have been set to satisfy public health authority specifications, e.g. for diabetic retinopathy [34]. Some rejection methods additionally weigh the chance of algorithmic error against that of human error [41].

When instead using the propensity score for rejection, a simple policy is to specify $\eta_0$ and reject points that do not satisfy eq. (4) with $\eta = \eta_0$. More sophisticated standard guidelines were proposed by Caliendo & Kopeinig [4]. These methods only account for the uncertainty about $\mathrm{CATE}(\boldsymbol{x})$ that is due to limited overlap and do not consider that uncertainty is also modulated by the availability of data on similar individuals (as well as the noise in this data).

## 4 Adapting neural causal models for covariate shift

### 4.1 Parameter uncertainty

To obtain the epistemic uncertainty in the CATE, we must infer the parameter uncertainty distribution conditioned on the training data $p(\boldsymbol{\omega}_0, \boldsymbol{\omega}_1|\mathcal{D})$, which defines the distribution of each network $\mu^{\boldsymbol{\omega}_0}(\cdot), \mu^{\boldsymbol{\omega}_1}(\cdot)$, conditioned on $\mathcal{D}$. There exists a large suite of methods we can leverage for this task, surveyed in Gal [14]. Here, we use MC Dropout [15] because of its high scalability [56], ease of implementation, and state-of-the-art performance [12]. However, our contributions are compatible

with other approximate inference methods. We can adapt almost all neural causal inference methods we know. CEVAE, however, [36], is more complicated and will be addressed in the next section.

MC Dropout is a simple change to existing methods. Gal & Ghahramani [15] showed that we can simply add dropout [52] with L2 regularization in each of $\boldsymbol{\omega}_0, \boldsymbol{\omega}_1$ during training and then sample from the same dropout distribution at test time to get samples from $q(\boldsymbol{\omega}_0, \boldsymbol{\omega}_1|\mathcal{D})$. With tuning of the dropout probability, this is equivalent to sampling from a Bernoulli approximate posterior $q(\boldsymbol{\omega}_0, \boldsymbol{\omega}_1|\mathcal{D})$ (with standard Gaussian prior). MC Dropout has been used in various applications [60, 38, 28].

## 4.2 Bayesian CEVAE

The Causal Effect Variational Autoencoder (CEVAE, Louizos et al. [36]) was introduced as a means to relax the common assumption that the data points $\boldsymbol{x}_i$ contain accurate measurements of all confounders – instead, it assumes that the observed $\boldsymbol{x}_i$ are a noisy transformation of some true confounders $\boldsymbol{z}_i$, whose conditional distribution can nonetheless be recovered. To do so, CEVAE encodes each observation $(\boldsymbol{x}_i, t_i, y_i) \in \mathcal{D}$, into a distribution over latent confounders $\boldsymbol{z}_i$ and reconstructs the entire observation with a decoder network. For each possible value of $t \in \{0, 1\}$, there is a separate branch of the model. For each branch $j$, the encoder has an auxiliary distribution $q(y_i|\boldsymbol{x}_i, t = j)$ to approximate the posterior $q(\mathbf{z}_i|\mathbf{x}_i, y_i, t = j)$ at test time. It additionally has a single auxiliary distribution $q(t_i|\mathbf{x}_i)$ which generates $t_i$. See Figure 2 in [36] for an illustration. The decoder reconstructs the entire observation, so it learns the three components of $p(\mathbf{x}_i, t_i, y_i|\mathbf{z}_i) = p(t_i|\mathbf{z}_i) \, p(y_i|t_i, \mathbf{z}_i) \, p(\mathbf{x}_i|\mathbf{z}_i)$. We will omit the parameters of these distributions to ease our notation. The encoder parameters are summarized as $\psi$ and the decoder parameters as $\boldsymbol{\omega}$.

If the treatment and outcome were known at test time, the training objective (ELBO) would be

$$\mathcal{L} = \sum_{i=1}^{N} \mathbb{E}_{q(\mathbf{z}_i|\mathbf{x}_i, t_i, y_i)} \big[ \log p\left(\mathbf{x}_i, t_i|\mathbf{z}_i\right) + \log p\left(y_i|t_i, \mathbf{z}_i\right) \big] - KL(q(\mathbf{z}_i|\mathbf{x}_i, t_i, y_i) \,||\, p(\mathbf{z}_i)) \quad (7)$$

where $KL$ is the Kullback-Leibler divergence. However, $t_i$ and $y_i$ need to be predicted at test time, so CEVAE learns the two additional distributions by using the objective

$$\mathcal{F} = \mathcal{L} + \sum_{i=1}^{N} (\log q(t_i = t_i^*|\mathbf{x}_i) + \log q(y_i = y_i^*|\mathbf{x}_i, t_i^*)), \quad (8)$$

where a star indicates that the variable is only observed at training time. At test time, we calculate the CATE so $t_i$ is set to 0 and 1 for the corresponding branch and $y_i$ is sampled both times.

Although the encoder performs Bayesian inference to infer $\boldsymbol{z}_i$, CEVAE does not model epistemic uncertainty because the decoder lacks a distribution over $\boldsymbol{\omega}$. The recently introduced Bayesian Variational Autoencoder [9] attempts to model such epistemic uncertainty in VAEs using MCMC sampling. We adapt their model for causal inference by inferring an approximate posterior $q(\boldsymbol{\omega}|\mathcal{D})$. In practice, this is again a simple change if we use Monte Carlo (MC) Dropout in the decoder[2]. This is implemented by adding dropout layers to the decoder and adding a term $KL(q(\boldsymbol{\omega}|\mathcal{D})||p(\boldsymbol{\omega}))$ to eq. (8), where $p(\boldsymbol{\omega})$ is standard Gaussian. Furthermore, the expectation in eq. (7) now goes over the *joint* posterior $q(\mathbf{z}_i|\mathbf{x}_i, t_i, y_i)q(\boldsymbol{\omega}|\mathcal{D})$ by performing stochastic forward passes with Dropout 'turned on'. Likewise, the joint posterior is used in the right term of eq. (6).

**Negative sampling for non-overlap.** *Negative sampling* is a powerful method for modeling uncertainty under a covariate shift by adding loss terms that penalize confident predictions on inputs sampled outside the training distribution [54, 33, 18, 19, 44]. However, it is usually intractable because the $\boldsymbol{x}$ input space is high dimensional. Our insight is that it becomes tractable for non-overlap, because the OoD inputs are created by simply flipping $t$ on the in-distribution inputs $\{(\boldsymbol{x}_i, t_i)\}$ to create the new inputs $\{(\boldsymbol{x}_i, t_i' = 1 - t_i)\}$. Our negative sampling is implemented by mapping each $(\boldsymbol{x}_i, y_i, t_i) \in \mathcal{D}$ through *both* branches of the encoder. On the *counterfactual* branch, where $t_i' = 1 - t_i$, we only minimize the KL divergence from the posterior $q(\boldsymbol{z}|\boldsymbol{x}_{\text{test}}, t = 0)$ to $p(\boldsymbol{z})$, but none of the other terms in eq. (8). This is to encode that we have no information on the counterfactual prediction. In appendix C.1 we study negative sampling and demonstrate improved uncertainty.

# 5   Related work

Epistemic uncertainty is modeled out-of-the-box by non-parametric Bayesian methods such as Gaussian Processes (GPs) [42] and Bayesian Additive Regression Trees (BART) [5]. Various non-parametric models have been applied to causal inference [2, 5, 59, 23, 58]. However, recent state-of-the-art results for high-dimensional data have been dominated by neural network approaches [27, 36, 3, 48]. Since these do not incorporate epistemic uncertainty out-of-the-box, our extensions are meant to fill this gap in the literature.

Causal effects are usually estimated after discarding/rejecting points that violate overlap, using the estimated propensity score [6, 20, 13, 48, 30, 7]. This process is cumbersome, and results are often sensitive to a large number of ad hoc choices [22] which can be avoided with our methods. Hill & Su [21] proposed alternative heuristics for discarding by using the epistemic uncertainty provided by BART on low dimensional data, but focuses on learning the ATE, the average treatment effect over the training set, and neither uses uncertainty in CATE nor ATE.

In addtion to violations of overlap, we also address CATE estimation for *test* data. Test data introduces the possibility of covariate shift away from $p(\boldsymbol{x})$, which has been studied outside the causal inference literature [40, 35, 53, 50]. In both cases, we may wish to reject $\boldsymbol{x}$, e.g. to consult a human expert instead of making a possibly false treatment recommendation. To our knowledge, there has been no comparison of rejection methods for CATE inference.

# 6   Experiments

In this section, we show empirical evidence for the following claims: that our uncertainty aware methods are robust both to violations of the overlap assumption and a failure mode of propensity based trimming (6.1); that they indicate high uncertainty when covariate shifts occur between training and test distributions (6.2); and that they yield lower CATE estimation errors while rejecting fewer points than propensity based trimming (6.2). In the process, we introduce a new, high-dimensional, individual-level causal effect prediction benchmark dataset called CEMNIST to demonstrate robustness to overlap and propensity failure (6.1). Finally, we introduce a modification to the IHDP causal inference benchmark to explore covariate shift (6.2).

We evaluate our methods by considering *treatment recommendations*. A simple treatment recommendation strategy assigns $t = 1$ if the predicted $\widehat{\text{CATE}}(x_i)$ is positive, and $t = 0$ if negative. As stated in section 3.2, insufficient knowledge about an individual and high costs due to error necessitate informed *rejection policies* to formalize when a recommendation should be withheld.

We compare four rejection policies: *epistemic uncertainty* using $\text{Var}[\widehat{\text{CATE}}^{\boldsymbol{\omega}_{0/1}}(\boldsymbol{x})]$, *propensity quantiles*, *propensity trimming* [4] and *random* (implementation details of each policy are given in Appendix A.4). Policies are ranked according to the proportion of incorrect recommendations made, given a fixed rate ($r_{\text{rej}}$) of withheld recommendations. This corresponds to assigning a cost of 1 to making an incorrect prediction and a cost of 0 for either making a correct recommendation or withholding an automated recommendation and deferring the decision to a human expert instead. We also report the Precision in Estimation of Heterogenous Treatment Effect (PEHE) [23, 47] over the non-rejected subset. The mean and standard error of each metric is reported over a dataset-dependent number of training runs.

We evaluate and compare each rejection policy using several uncertainty aware CATE estimators. The estimators are Bayesian versions of CEVAE [36], TARNet, CFR-MMD [47], Dragonnet [48], and a deep T-Learner. Each model is augmented by introducing Bayesian parameter uncertainty and by predicting a distribution over model outputs. For imaging experiments, a two-layer CNN encoder is added to each model. Details for each model are given in Appendix B. In the result tables, each model's name is prefixed with a "B" for "Bayesian". We also compare to Bayesian Additive Regression Trees (BART) [23].

## 6.1   Using uncertainty when overlap is violated

**Causal effect MNIST (CEMNIST).** We introduce the CEMNIST dataset using hand-written digits from the MNIST dataset [32] to demonstrate that our uncertainty measures capture non-overlap on high-dimensional data and that they are robust to a failure mode of propensity score rejection.

Table 1: **CEMNIST-Overlap** Description of "Causal effect MNIST" dataset.

| Digit(s) | $p(\boldsymbol{x})$ | $p(t=1\|\boldsymbol{x})$ | $p(y=1\|\boldsymbol{x}, t=0)$ | $p(y=1\|\boldsymbol{x}, t=1)$ | CATE |
|---|---|---|---|---|---|
| 9 | 0.5 | 1/9 | 1 | 0 | $-1$ |
| 2 | 0.5/9 | 1 | 0 | 1 | 1 |
| other odds | 0.5/9 | 0.5 | 1 | 0 | $-1$ |
| other evens | 0.5/9 | 0.5 | 0 | 1 | 1 |

Table 1 depicts the data generating process for CEMNIST. In expectation, half of the samples in a generated dataset will be nines, and even though the propensity for treating a nine is relatively low, there are still on average twice as many treated nines as there are samples of other treated digits (except for twos). Therefore, it is reasonable to expect that the CATE can be estimated most accurately for nines. For twos, there is strict non-overlap. Therefore, the CATE cannot be estimated accurately. For the remaining digits, the CATE estimate should be less confident than for nines because there are fewer examples during training, but more confident than for twos because there are both treated and untreated training examples.

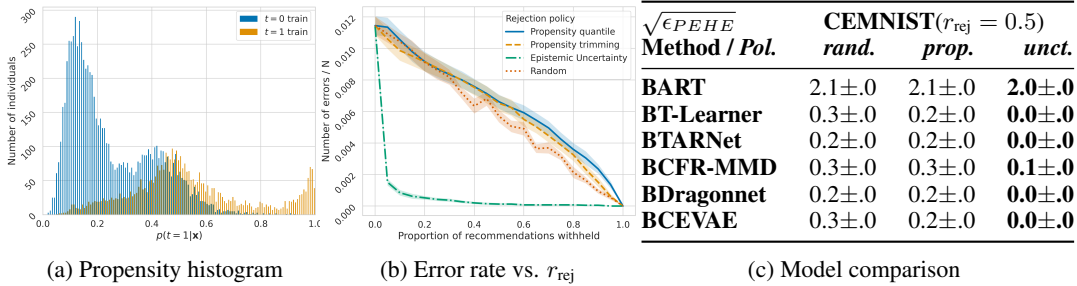

(a) Propensity histogram     (b) Error rate vs. $r_{\text{rej}}$     (c) Model comparison

| $\sqrt{\epsilon_{PEHE}}$ | CEMNIST($r_{\text{rej}} = 0.5$) | | |
|---|---|---|---|
| Method / *Pol.* | *rand.* | *prop.* | *unct.* |
| **BART** | 2.1±.0 | 2.1±.0 | **2.0±.0** |
| **BT-Learner** | 0.3±.0 | 0.2±.0 | **0.0±.0** |
| **BTARNet** | 0.2±.0 | 0.2±.0 | **0.0±.0** |
| **BCFR-MMD** | 0.3±.0 | 0.3±.0 | **0.1±.0** |
| **BDragonnet** | 0.2±.0 | 0.2±.0 | **0.0±.0** |
| **BCEVAE** | 0.3±.0 | 0.2±.0 | **0.0±.0** |

Figure 2: **CEMNIST** evaluation. (a) Histogram of estimated propensity scores. Untreated nines account for the peaks on the left side. (b) Error rate for different rejection policies as we vary the rejection rate. (c) $\sqrt{\epsilon_{PEHE}}$ for different models at a fixed rejection rate $r_{\text{rej}} = 0.5$. Compared are the policies *random*, *propensity trimming*, and *epistemic uncertainty*.

This experimental setup is chosen to demonstrate where the *propensity* based rejection policies can be inappropriate for the prediction of individual-level causal effects. Figure 2a shows the histogram over training set predictions for a deep propensity model on a realization of the CEMNIST dataset. A data scientist following the trimming paradigm [4] would be justified in choosing a lower threshold around 0.05 and an upper threshold around 0.75. The upper threshold would properly reject twos, but the lower threshold would start rejecting nines, which represent the population that the CATE estimator can be most confident about. Therefore, rejection choices can be worse than random.

Figure 2b shows that the recommendation-error-rate is significantly lower for the *epistemic uncertainty* policy (green dash-dot) than for both the *random* baseline policy (red dot) and the *propensity* based policies (orange dash and blue solid). BT-Learner is used for this plot. These results hold across a range of other SOTA CATE estimators for the $\sqrt{\epsilon_{PEHE}}$, as shown in figure 2c, and in Appendix C.1. Details on the protocol generating these results are in Appendix A.1.

Table 2: Comparing *epistemic uncertainty*, *propensity trimming*, and *random* rejection policies for IHDP, IHDP Covariate Shift, and ACIC 2016 and with uncertainty-equipped SOTA models. 50% or 10% of examples set to be rejected and errors are reported on the remaining test-set recommendations. *Epistemic uncertainty* policy leads to the lowest errors in CATE estimates (in bold).

| $\sqrt{\epsilon_{PEHE}}$ | IHDP ($r_{\text{rej}} = 0.1$) | | | IHDP Cov. ($r_{\textbf{rej}} = 0.5$) | | | ACIC 2016 ($r_{\text{rej}} = 0.1$) | | |
|---|---|---|---|---|---|---|---|---|---|
| Method / *Pol.* | *rand.* | *prop.* | *unct.* | *rand.* | *prop.* | *unct.* | *rand.* | *prop.* | *unct.* |
| **BART** | 1.9±.2 | 1.9±.2 | **1.6±.1** | 2.6±.2 | 2.7±.3 | **1.8±.2** | 1.3±.1 | 1.2±.1 | **0.9±.1** |
| **BT-Learner** | 1.0±.0 | 0.9±.0 | **0.7±.0** | 2.3±.2 | 2.3±.2 | **1.3±.1** | 2.1±.1 | 2.0±.1 | **1.5±.1** |
| **BTARNet** | 1.1±.0 | 1.0±.0 | **0.8±.0** | 2.2±.3 | 2.0±.3 | **1.2±.1** | 1.8±.1 | 1.7±.1 | **1.2±.1** |
| **BCFR-MMD** | 1.3±.1 | 1.3±.1 | **0.9±.0** | 2.5±.2 | 2.4±.3 | **1.7±.2** | 2.3±.2 | 2.1±.1 | **1.7±.1** |
| **BDragonnet** | 1.5±.1 | 1.4±.1 | **1.1±.0** | 2.4±.3 | 2.2±.3 | **1.3±.2** | 1.9±.1 | 1.8±.1 | **1.3±.1** |
| **BCEVAE** | 1.8±.1 | 1.9±.1 | **1.5±.1** | 2.5±.2 | 2.4±.3 | **1.7±.1** | 3.3±.2 | 3.2±.2 | **2.9±.1** |

## 6.2 Uncertainty under covariate shift

**Infant Health and Development Program (IHDP).** When deploying a machine learning system, we must often deal with a test distribution of $x$ which is different from the training distribution $p(x)$. We induce a covariate shift in the semi-synthetic dataset IHDP [23, 47] by excluding instances *from the training* set for which the mother is unmarried. Mother's marital status is chosen because it has a balanced frequency of $0.52 \pm 0.00$; furthermore, it has a mild association with the treatment as indicated by a log odds ratio of $2.22 \pm 0.01$; and most importantly, there is evidence of a simple distribution shift, indicated by a predictive accuracy of $0.75 \pm 0.00$ for marital status using a logistic regression model over the remaining covariates. We comment on the ethical implications of this experimental set-up, describe IHDP, and explain the experimental protocol in Appendix A.2.

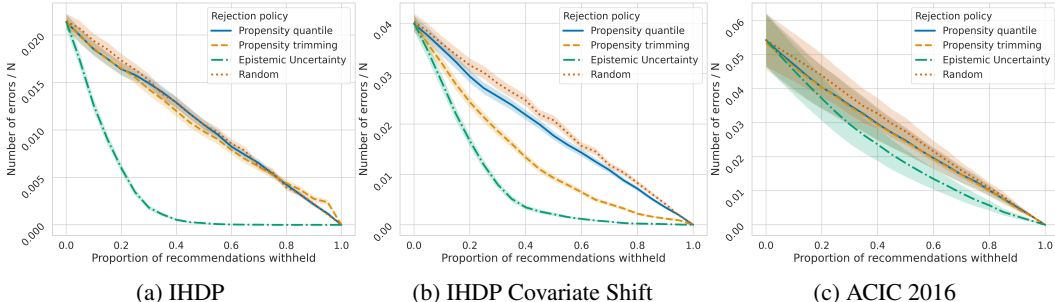

(a) IHDP        (b) IHDP Covariate Shift        (c) ACIC 2016

Figure 3: Uncertainty based rejection policies yield significantly lower error rates while withholding fewer recommendations than propensity policies, on IHDP, IHDP Cov., and ACIC 2016.

We report the mean and standard error in recommendation-error-rates and $\sqrt{\epsilon_{PEHE}}$ over 1000 realizations of the IHDP Covariate-Shift dataset to evaluate each policy by computing each metric over the test set (both sub-populations included). We sweep $r_{\text{rej}}$ from 0.0 to 1.0 in increments of 0.05. Figure 3b shows, for the BT-Learner, that the *epistemic uncertainty* (green dash-dot) policy significantly outperforms the uncertainty-oblivious policies across the whole range of rejection rates, and we show in Appendix C that this trend holds across all models. The middle section of table 2 supports this claim by reporting the $\sqrt{\epsilon_{PEHE}}$ for each model at $r_{\text{rej}} = 0.5$; the approximate frequency of the excluded population. Every model class shows improved rejection performance. However, comparisons between model classes are not necessarily appropriate since some models target different scenarios, for example, CEVAE targets *non*-synthetic data where confounders $z$ aren't directly observed, and it is known to underperform on IHDP [36].

We report results for the unaltered IHDP dataset in figure 3a and the l.h.s. of table 2. This supports that uncertainty rejection is more *data-efficient*, i.e., errors are lower while rejecting less. This is further supported by the results on ACIC 2016 [11] (figure 3c and the r.h.s. of table 2). The preceding results can be reproduced using publicly available code[3].

## 7 Conclusions

Observational data often violates the crucial overlap assumption, especially when the data is high-dimensional [8]. When these violations occur, causal inference can be difficult or impossible, and ideally, a good causal model should communicate this failure to the user. However, the only current approach for identifying these failures in deep models is via the propensity score. We develop here a principled approach to modeling outcome uncertainty in individual-level causal effect estimates, leading to more accurate identification of cases where we cannot expect accurate predictions, while the propensity score approach can be both over- and under-conservative. We further show that the same uncertainty modeling approach we developed can be usefully applied to predicting causal effects under covariate shift. More generally, since causal inference is often needed in high-stakes domains such as medicine, we believe it is crucial to effectively communicate uncertainty and refrain from providing ill-conceived predictions.

## 8 Broader impact

Here, we highlight a set of beneficial and potentially alarming application scenarios. We are excited about our methods to contribute to ongoing efforts to create neural treatment recommendation systems that can be safely used in medical settings. Safety, along with performance, is a major roadblock for this application. In regions where medical care is scarce, it may be especially likely that systems will be deployed despite limited safety, leading to potentially harmful recommendations. In regions with more universal medical care, individual-based recommendations could improve health outcomes, but systems are unlikely to be deployed when they are not deemed safe.

Massive observational datasets are available to consumer-facing online businesses such as social networks, and to some governments. For example, standard inference approaches are limited for recommendation systems on social media sites because a user's decision to follow a recommendation (the treatment) is confounded by the user's attributes (and even the user-base itself can be biased by the recommendation algorithm's choices) [46]. Causal approaches are therefore advantageous. Observational datasets are typically high-dimensional, and therefore likely to suffer from severe overlap violations, making the data unusable for causal inference, or implying the need for cumbersome preprocessing. As our methods enable working directly with such data, they might enable the owners of these datasets to construct causal models of *individual* human behavior, and use these to manipulate attitudes and behavior. Examples of such manipulation include affecting voting and purchasing choices.

## 9 Acknowledgements

We would like to thank Lisa Schut, Clare Lyle, and all anonymous reviewers for their time, effort, and valuable feedback. S.M. is funded by the Oxford-DeepMind Graduate Scholarship. U.S. was partially supported by the Israel Science Foundation (grant No. 1950/19).

## Footnotes

[2]We do not treat the parameters $\psi$ of the encoder distributions as random variables. This is because the encoder does not infer $\boldsymbol{z}$ directly. Instead, it parameterizes the *parameters* $\mu(\boldsymbol{z}), \Sigma(\boldsymbol{z})$ of a Gaussian posterior over $\boldsymbol{z}$ (see eq. (5) in Louizos et al. [36] for details). These parameters specify the uncertainty over $\boldsymbol{z}$ themselves.

[3]Available at: `https://github.com/OATML/ucate`

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
