[Supplementary Material · supplementary.pdf]

# Appendix A Datasets

## A.1 CEMNIST

Table 3: **CEMNIST-Overlap** Description of "Causal effect MNIST" dataset.

| Digit(s) | Number of train samples | Number treated | $Y^0$ | $Y^1$ | CATE |
|---|---|---|---|---|---|
| 9 | 6000 | $\approx 666$ | 1 | 0 | $-1$ |
| 2 | $\approx 666$ | $\approx 666$ | 0 | 1 | 1 |
| other odds | $\approx 666$ each | $\approx 333$ each | 1 | 0 | $-1$ |
| other evens | $\approx 666$ each | $\approx 333$ each | 0 | 1 | 1 |

The original MNIST image dataset contains a training set of size 60000 and a test set of size 10000, where each digit class 0-9 represents 10% of points. We use a subset of the training data, shown in Table 3. Similarly, we use a subset of the test set, with the same proportion for each digit as in the training set (and the same proportion of treated points). The variables $Y^1$, $Y^0$ are deterministic as shown in Table 3. Some numbers in Table 3 are approximate because they are generated according to the probabilities in Table 1.

The dataset serves two purposes. First, it illustrates why the standard practice of rejecting points with propensity scores close to $0$ or $1$ can be worse than rejecting randomly. The digit 9 has the most data making it easy to predict the CATE, but its propensity score is only $0.1$, so that 9s will be rejected early. It might be a common situation in practice that a sub-population represents the majority of the data and therefore its CATE is easy to estimate. Second, the digit 2 suffers from strict non-overlap (propensity score of $1$). It should be the first digit class to be rejected by any method since its CATE cannot be estimated. When increasing the rejected proportion, digits other than 9 should subsequently be rejected as only $334$ and $333$ examples are observed for their treatment and control groups respectively. However, propensity-based rejection is likely to retain these sub-populations because their propensity score is $0.5$.

We repeated the CEMNIST experiment 20 times, each time generating a new dataset with a different random initialization for each model. Note that this is a single dataset, unlike other causal inference benchmarks, so it is only suited for CATE estimation, not ATE estimation.

## A.2 IHDP

Hill [23] introduced a causal inference dataset based on the The Infant Health Development Program (IHDP), a randomized experiment that assessed the impact of specialist home visits on children's performance in cognitive tests. Real covariates and treatments related to each participant are used in the IHDP dataset. However, outcomes are simulated based on covariates and treatment, making this dataset semi-synthetic. Covariates were made different between the treatment and control groups by removing units with non-white mothers from the treated population. There are 747 units in the dataset (139 treated, 608 control), with 25 covariates related to the children and their mothers. Following Shalit et al. [47], Hill [23], we use the simulated outcome implemented as setting "A" in the NPCI package [10] and we use the noiseless/expected outcome to compute the ground truth CATE. The IHDP dataset is available for download at `https://www.fredjo.com/`.

We run the experiment according to the protocol described in [47]: we run 1000 repetitions of the experiment, where each test set has 75 points and the remaining 672 available points are split 70% to 30% for training and validation. The ground truth outcomes are normalized to a mean of $0$ and standard deviation of $1$ over the training set. For evaluation, each model's predictions are unnormalized to calculate the PEHE.

**IHDP Covariate Shift.** As previously mentioned, we selected a variable (marital status of mother) and exclude datapoints where the mother was unmarried from training (while leaving the test set unaltered). We selected this feature for three reasons: it is active in roughly 50% of data points, the distributions of the remaining covariates were distinct based on a T-SNE visualization [37], and the feature is only marginally correlated with treatment (which ensures that we study the impact of covariate shift, not unobserved confounding). The feature is hidden to the models to make the detection of covariate shift non-trivial, and to induce a more realistic scenario where latent factors are often unaccounted for in observational data.

Marital status may be considered a sensitive socio-economic factor. We do not intend the experiment to be politically insensitive, rather that it emphasizes the problem of demographic exclusion in observational data due to issues such as historical bias, along with the danger of making confident but uninformed predictions when demographic exclusion is latent. Omitting these variables can lead to subpar model performance – particularly for members of a socio-economic minority.

### A.3   ACIC 2016

Dorie et al. [11] introduced a dataset named after the 2016 Atlantic Causal Inference Conference (ACIC) where it was used for a competition. ACIC is a collection of semi-synthetic datasets whose covariates are taken from a large study conducted on pregnant women and their children to identifying causal factors leading to developmental disorders [39]. There are 4802 observations and 58 covariates. Outcomes and treatments are simulated, as in IHDP, according to different data-generating process for each dataset. We chose this dataset instead of the 2018 ACIC challenge [51] because the latter is aimed at only ATE estimation and the CATE is equal for each observation in most datasets.

### A.4   Evaluation metrics

We evaluate our methods by considering *treatment recommendations*. A simplified recommendation strategy for an individual-level treatment of a unit with covariates $x_i$ is to recommend $t = 1$ if the predicted $\text{CATE}(x_i)$ is positive, and $t = 0$ if negative. However, if there is insufficient knowledge about the CATE an individual, and a high cost associated with making errors, it may be preferable to withhold the recommendation, and e.g. refer the case for further scrutiny. It is therefore important to have an informed *rejection policy* for a treatment assigned based on a given CATE estimator.

To evaluate a rejection policy for a CATE estimator we assign a cost of 1 to making incorrect predictions and a cost of 0 for making a correct recommendation. At a fixed number of rejections, the utility of a policy can be defined as the inverse of the total number of erroneous recommendations made, i.e., if a policy can correctly identify the model's mistakes and refer such patients to a human expert then it should have a higher utility.

**Rejection policies** We introduce two rejection policies based on the epistemic and predictive uncertainty estimates of an uncertainty aware CATE estimator. Both policies opt to reject if the uncertainty estimate is greater than a threshold that rejects a given proportion of the training data $r_{\text{reject}}$. The training data is used since there may not be a large enough test set in practice. For all policies, we determine thresholds on the training set to simulate a real-world individual-level recommendation scenario. The *epistemic uncertainty* policy uses a sample-based estimator of the uncertainty in CATE (second *r.h.s.* term in eq. (6)) given by

$$\widehat{Var}_{epi}[\mu^1(\boldsymbol{x}_i) - \mu^0(\boldsymbol{x}_i)] := \frac{1}{M} \sum_{j=1}^{M} \left( \mu^{\hat{\boldsymbol{\omega}}_j^1}(\boldsymbol{x}_i) - \mu^{\hat{\boldsymbol{\omega}}_j^0}(\boldsymbol{x}_i) \right)^2 - \left( \frac{1}{M} \sum_{j=1}^{M} \mu^{\hat{\boldsymbol{\omega}}_j^1}(\boldsymbol{x}_i) - \mu^{\hat{\boldsymbol{\omega}}_j^0}(\boldsymbol{x}_i) \right)^2 ,$$

(9)

where $M$ Monte Carlo samples are taken from each of $q(\boldsymbol{\omega}_0, \boldsymbol{\omega}_1 | \mathcal{D})$. Note that, for the T-Learner, this posterior factorizes into two independent distributions $q(\boldsymbol{\omega}_0 | \mathcal{D}), q(\boldsymbol{\omega}_1 | \mathcal{D})$ because there are separate models for the outcome given treatment and no treatment. Furthermore, other models share parameters for $\mu^{\boldsymbol{\omega}_0}(\cdot), \mu^{\boldsymbol{\omega}_1}(\cdot)$ so the individual parameters in $\boldsymbol{\omega}_0, \boldsymbol{\omega}_1$ may overlap. The *predictive* uncertainty policy uses an estimator of eq. (6), $\widehat{Var}_{pred}[Y^1 - Y^0 | \boldsymbol{x}_i]$, which has the same functional form as in eq. (9), but instead of being over the difference in expected values $\mu^{\hat{\boldsymbol{\omega}}_j^t}(\boldsymbol{x}_i)$ of the output distribution it is over samples $y^{\hat{\boldsymbol{\omega}}_j^t}(\boldsymbol{x}_i)$ of the output distribution.

We compare the utility of these policies to a random rejection baseline and two policies based on propensity scores. The first propensity policy (*propensity quantiles*) finds a two sided threshold on the distribution of estimated propensity scores such that a proportion $(1. - r_{\text{reject}})$ of the training data is retained. The second policy (*propensity trimming*) implements a trimming algorithm following the guidelines proposed by Caliendo & Kopeinig [4].

## Appendix B   Models

We evaluate and compare each rejection policy using several uncertainty-aware CATE estimators. The estimators are the Bayesian versions of CEVAE [36], TARNet, CFR-MMD [47], Dragonnet [48], and a deep version of the T-Learner [47]. Each model is augmented by introducing Bayesian

parameter uncertainty and by predicting a distribution over model outputs. For image data, two convolutional bottom layers are added to each model.

Each model is augmented with Bayesian parameter uncertainty by adding dropout with a probability of 0.1 after each layer (0.5 for layers before the output layer), and setting weight decay penalties to be inversely proportional to the number of examples in the training dataset. At test time, uncertainty estimates are calculated over 100 MC samples.

For the Bayesian T-Learner we use two BNNs, each having 5 dense, 200 neuron, layers. Dropout is added after each dense layers, followed by ELU activation functions. A linear output layer is added to each network, with a sigmoid activation function if the target is binary. For image data, we add a 2-layer convolutional neural network module, with 32 and 64 filters per layer. Spatial dropout [55], and ELU activations follow each convolutional layer, and the output is flattened before being passed to the rest of the network. For image data, the Bayesian CEVAE decoder is modified by using a transposed convolution block for the part of the decoder that models $p(\boldsymbol{x}|\boldsymbol{z})$. For the propensity policies, we use a propensity model that has the same form as a single branch of the Bayesian T-learner. The propensity model's L2 regularization is tuned for calibration as this is important for propensity models. We also experimented with a logistic regression model which performed worse.

Adam optimization [31] is used with a learning rate of 0.001 (On CEMNIST the learning rate for the BCEVAE is reduced to 0.0002), and we train each model for a maximum of 2000 epochs, using early stopping with a patience of 50.

Aside from these changes, model architectures, optimization strategies and loss weighting follow what is reported in their respective papers. More details can be seen in the attached code.

### B.1 Compute infrastructure

All neural network models were implemented in Tensorflow 2.2 [1], using Nvidia GPUs. BART was implemented using the dbarts R package, available at `https://cran.r-project.org/web/packages/dbarts/index.html`.

## Appendix C  Additional Results

### C.1  CEVAE negative sampling

Figure 4 illustrates the benefit of negative sampling for detecting examples in violation of overlap, or out-of-distribution. Negative sampling results in higher epistemic uncertainty measures and sharper transitions between in-distribution and out-of-distribution regions.

Table 4 and figure 5 compare the BCEVAE model when trained with and without negative sampling on the CEMNIST dataset.

Table 4: Comparing BCEVAE trained with and without-negative sampling on CEMNIST. 50% of examples set to be rejected and errors are reported on the remaining test-set recommendations. *Epistemic uncertainty* policy leads to the lowest errors in CATE estimates (in bold).

| Method / *Pol.* | $\sqrt{\epsilon_{PEHE}}$ ($r_{\text{rej}} = 0.5$) | | | Rec. Err. ($r_{\text{rej}} = 0.5$) | | |
| | *rand.* | *prop.* | *unct.* | *rand.* | *prop.* | *unct.* |
|---|---|---|---|---|---|---|
| **Negative Sampling** | .295±.005 | .227±.007 | **.037±.009** | .010±.001 | .005±.001 | **.000±.000** |
| **No Negative Sampling** | .286±.005 | .226±.007 | **.033±.007** | .011±.001 | .007±.001 | **.000±.000** |

### C.2  IHDP

Table 5 shows the relative performance of the Bayesian models to the results reported in their respective papers for the IHDP dataset.

(a) Epistemic uncertainty $\mathcal{I}$ for Bayesian CEVAE trained with negative sampling

(b) Epistemic uncertainty $\mathcal{I}$ for Bayesian CEVAE trained without negative sampling

Figure 4: Comparing epistemic uncertainty measures for Bayesian CEVAE trained (a) *with* negative sampling and (b) *without* negative sampling during training. Both models give appropriately low uncertainty measures for in-distribution examples (pink, no shading), and appropriately high uncertainty measures for out-of-distribution examples (dark-blue, dark shading). However, the model trained with negative sampling gives higher uncertainty measures and sharper transitions for non-overlap examples (light-blue, light shading). These properties are important as we propose to use the uncertainty measures to define policies of when to defer a treatment recommendation and instead seek out an expert opinion.

Table 5: Errors on unaltered IHDP, comparing to previously published results (given in the upper half). Note that BCEVAE (ours) outperforms CEVAE. $\epsilon_{ATE}$ is the squared error of the Average Treatment Effect. These results are only for completeness and do not contain the evidence for our main findings.

| | within-sample | | out-of-sample | |
|---|---|---|---|---|
| **Method** | $\sqrt{\epsilon_{PEHE}}$ | $\epsilon_{ATE}$ | $\sqrt{\epsilon_{PEHE}}$ | $\epsilon_{ATE}$ |
| **OLS-2** | 2.4±.1 | .14±.01 | 2.5±.1 | .31±.02 |
| **BART** | 2.1±.1 | .23±.01 | 2.3±.1 | .34±.02 |
| **BNN** | 2.2±.1 | .37±.03 | 2.1±.1 | .42±.03 |
| **GANITE** | 1.9±.4 | .43±.05 | 2.4±.4 | .49±.05 |
| **CEVAE** | 2.7±.1 | .34±.01 | 2.6±.1 | .46±.02 |
| **TARNet** | .88±.0 | .26±.01 | .95±.0 | .28±.01 |
| **CFR-MMD** | .73±.0 | .3±.01 | .78±.0 | .31±.01 |
| **Dragonnet** | | .14±.01 | | .20±.01 |
| **BT-Learner** | .95±.0 | .21±.01 | .88±.0 | .18±.01 |
| **BTARNet** | 1.1±.0 | .23±.01 | .96±.0 | .20±.01 |
| **BCFR-MMD** | 1.3±.1 | .29±.01 | 1.2±.1 | .26±.01 |
| **BDragonnet** | 1.5±.1 | .30±.01 | 1.3±.0 | .27±.01 |
| **BCEVAE** | 1.8±.1 | .47±.01 | 1.8±.1 | .50±.02 |

(a) **Negative Sampling**: Errors

(b) **Negative Sampling**: $\sqrt{\epsilon_{PEHE}}$

(c) **No Negative Sampling**: Errors

(d) **No Negative Sampling**: $\sqrt{\epsilon_{PEHE}}$

Figure 5: **CEMNIST BCEVAE** with and without negative sampling.