[Reviews · NeurIPS 2020]

Review 1

Summary and Contributions: This paper is about equipping estimators of individual treatment effects (or heterogeneous treatment effects) with an awareness of uncertainty. For example, on a new sample that doesn't resemble any training sample that received treatment, an ITE estimator should know that it doesn't know enough to predict an effect of intervention and defer. The paper first connects this uncertainty to problems of covariate shift and violations of overlap (some subgroups never receive treatment). The paper derives a notion of uncertainty for estimators of the conditional average treatment effect (CATE). Then, the paper proposes a simple modification to many neural network estimators that can estimate this uncertainty quantity using dropout. The paper demonstrates this plug-in uncertainty estimator idea for many SoTA methods to produce CATE and show that it overwhelmingly allows methods to recognize samples that violated overlap or don't resemble training data.

Strengths: + The motivation of the paper is strong. The introduction gives a compelling example of individual treatment effects in medical settings where it's important for an estimator to know that a new patient doesn't look like any sample it's been trained on, and defers prediction instead of e.g., confidently offering a highly spurious recommendation to treat the patient. This example makes it clear why this problem is important --- covariate shift between train and test sets or violations of overlap within the training data are common. It's a valuable contribution to the causal inference community to tackle the problem of imbuing CATE estimators with uncertainty estimates. + The paper is clear and well-written. I found the notation easy to understand and the technical ideas easy to follow. Moreover, the presentation is high quality. For example, the figures in the paper like Figure 1 are helpful for illustrating the ideas. + The empirical studies are compelling, especially on IHDP. Five different well-known SoTA methods for CATE estimation applied to IHDP have been modified with the paper's proposed uncertainty-awareness scheme, and the results are convincing. Moreover, I found the evaluation method and descriptions of the experiments clear and easy to follow.

Weaknesses: I emphasize that the weaknesses of this paper are minor, in my view. - There is occasional exposition that I don't feel precisely captures causal inference. For example, in the introduction, the authors say "However, the price we pay for using observational data is lower certainty in our causal estimates, due to the possibility of unmeasured confounding, and the measured and unmeasured differences between the populations who were subject to different treatments." This isn't quite right --- for example, under unmeasured confounding, the estimates can be arbitrarily biased even if they are low variance. The estimate is simply not identified. Another example is in section 2 after Equation 2. We consider the expected treatment effect not just because the outcome is random, but because the individual level treatment effect (inside the expectation) cannot be identified without parametric assumptions on the causal model. - I couldn't really follow the section on negative sampling. It wasn't clear what its role was in this particular method --- does it produce better calibrated estimates of the variance of CATE? Or does it do something else. This is part where exposition can be improved to help the reader follow better.

Correctness: The empirical methodology is rigorous and the strategy of evaluating recommendations made by the methods is useful. To the best of my knowledge, the derivation of key steps like Equation 7 look sound.

Clarity: The paper is overall well written.

Relation to Prior Work: The relation to prior work is clear to me. This paper aims to build upon existing CATE estimators by adapting them to estimate uncertainty. They use a well-known dropout estimator of variance, but adapt it for the causal inference setting. I think it's clear where this paper is situated in the context of related work.

Reproducibility: Yes

Additional Feedback: EDIT: Thanks to the authors for answering my questions in their response. I maintain my score and I encourage the authors to include the clarifications that they provided in the author response in the main paper. A minor point where I couldn't fully follow: Eqn 4 reviews one notion of uncertainty for regression tasks. Is Eqn 4 what motivates the derivation of the notion of CATE uncertainty in Eqn 7? I couldn't fully follow what exactly the point was of reviewing Eqn 4, and what motivated the choice of expanding the second moment Y^1 - Y^0 as the notion of uncertainty.


Review 2

Summary and Contributions: This paper looks at the problem of incorporating (epistemic) uncertainty estimates into methods for doing causal inference from observational data. Specifically, the paper is concerned with a setting where there is covariate shift: inferences made on one dataset may be applied to a related but dataset (e.g., administering medication to a patient whose medical characteristics differ from the population the drug has been tested on). In such settings, there is often a good deal of uncertainty as to what the treatment effect will be in the new population. The authors propose a way of quantifying this uncertainty and using it to identify decision makers to problematic cases when such examples are encountered.

Strengths: This work addresses the important problem of learning to "know when you don't know". Machine learning methods are often used to make out-of-distribution predictions, and, whether this population shift is explicitly acknowledged or not, methods will generate predictions but do not often indicate any uncertainty or confidence in these predictions. This paper presents a way quantifying that uncertainty under a variety of neural methods for estimating conditional average treatment effects. This can inform decision makers as to when a prediction shouldn't be trusted, or should at least be reviewed or double checked. The contribution appears novel and should be of interest to the NeurIPS community.

Weaknesses: The paper makes the assumption of ignorability---or that all possible confounds are observed. This is unlikely to hold in practice, which may limit applicability of the method. However the authors do show an extension to Causal Effect Variational Autoencoders, which have a weaker assumption that proxy variables exist that can be used to make inferences about any possible confounds.

Correctness: The claims and methods appear to be correct.

Clarity: Yes, the paper is well written.

Relation to Prior Work: The related work section does a good job of covering related work, but might also include this recent work on when to delegate to humans vs. algorithms: https://arxiv.org/abs/1903.12220

Reproducibility: Yes

Additional Feedback: All of the results in the paper seem to sweep over some threshold at which predictions are considered too uncertain, showing that at the same number of examples withheld, their method outperforms others. In practice, however, once will have to pick a threshold and stick with it. Can the authors offer any further insight into where or how this threshold should be set? AFTER READING THE AUTHORS' RESPONSE: Thanks to the authors for their thoughtful response. I maintain my original score and recommend the paper for acceptance. It would be great if the authors could include the comments on domain-specific concerns around setting thresholds from their response to reviewers in the main text.


Review 3

Summary and Contributions: This paper propose using model uncertainty to build a better rejection classifier to address the issue of unreliable causal effect estimation when the positivity assumption is violated. To achieve this, authors propose augment SOTA causal models with standard uncertainty method (MC Dropout), and use the estimated epistemic uncertainty to build a rejection classifier. Authors illustrated how MC Dropout can be applied to standard causal models (CEVAE, TARNet etc), and how to further improve the performance via negative sampling. On two semi-synthetic image and medical datasets, authors showed that the proposed method significantly outperforms the standard approach (propensity trimming).

Strengths: This paper illustrates a nice use case of improving causal effect estimate using model uncertainty, and illustrated clear improvement over the standard approaches in a realistic setting where the positivity assumption does not hold. The center claim of the paper is about the advantage of model uncertainty over the classic approach (propensity trimming) in handling positivity violation. The method is intuitively sensible and is nicely illustrated in the Sections 2.1 and 6.1. I find this paper to be of significant conceptual novelty, and is relevant to NeurIPS audience who are interested in causality and uncertainty quantification.

Weaknesses: Minor Issues: - Authors should add more detail to section 3 , after introducing how to estimate model uncertainty, authors should consider adding a new section 3.2. addressing how the model uncertainty can be used as a better replacement of the propensity trimming (e.g., use your description about rejection policy in section A.4.). - How can the epistemic uncertainty estimation formula (7) be applied to CEVAE, for which both w and z are random? Please consider adding explanation to Section 4.2. - In the experiments, the predictive uncertainty policy and the epistemic uncertainty policy appear to behave very similarly? Are there data / model scenario that we would prefer one over the other? Please consider adding comment regarding this issue, or discuss it in the conclusion section.

Correctness: Yes, the empirical evaluation appears to be sensible.

Clarity: The paper is missing some important technical details (see 'Weakness'). However it can be addressed by re-organizing materials from Appendix to main section, .

Relation to Prior Work: Yes, it is discussed in the Related work section.

Reproducibility: Yes

Additional Feedback: (Posted after author rebuttal). Thank authors for the response. I believe my concerns are sufficiently addressed and expect the author to make the promised change in the final paper. I maintain my original score of 7.


Review 4

Summary and Contributions: This paper introduces an individual-level causal estimation method which handles “no-overlap” situations and covariate shift. The authors analyze how covariate shift and violations of overlap are related to real-world applications when learning conditional average treatment effects from data.

Strengths: I find the paper clearly well written and very well presented. I find the idea of modeling outcome uncertainty in individual-level causal effect estimates very interesting as well as the uncertainty modeling is applied to predicting causal effects under covariate shift. It also conveys the main idea clearly. It’s an interesting combination of modeling uncertainty and CEVAE to tackle the no-overlap and covariate shift problems.

Weaknesses: Even though the paper handles an interesting setting and the proposed technical solutions look reasonable, the idea seems to be pretty incremental as it stacks multiple existing techniques without many innovations. In general, it was easy to read the paper, but I found some grammatical errors in some sections, which is trivial.

Correctness: The technical solutions presented in the paper look reasonable. The paper clearly discuss the connections to the pre-existing work by addressing their limitations and how they tackle the existing problems behind them.

Clarity: The paper is well-written.

Relation to Prior Work: The paper clearly discuss the connections to the pre-existing work by addressing their limitations and how they tackle the existing problems behind them.

Reproducibility: Yes

Additional Feedback: I believe the core value behind this work is an effort to handle no-overlap situation and covariate shift by using the combination of modeling uncertainty and causal effect variational auto-encoder. Yet, it’s a bit hard to say their proposed methodology is quite novel in terms of technicality. Can authors provide more idea with this? [Additional comments] Thanks to the authors for answering my questions in their response.

[Author Response · NeurIPS 2020]

We thank reviewers for their useful feedback. We are encouraged that all reviewers recognize the relevance and
importance of being uncertain when overlap does not hold or when covariate shift has occurred, and, moreover, by
recognizing our "valuable" (R1), "novel" (R2) and "interesting" (R4) contributions, the "significant conceptual novelty"
(R3) of our methods, and that our estimators "overwhelmingly allow methods to recognize samples that violate overlap
or don't resemble training data" (R1). We are pleased that reviewers find our empirical results convincing, that they show
significant improvement over baselines (R3) and are carried out through realistic (R3), rigorous (R1), and reproducible
experiments. Finally, we appreciate that the majority of concerns are given as suggestions for improvement and we
address the reviewers' insightful comments (noting that R2 understood the paper well–despite their low confidence
score) in the following.

**R1 "There is occasional exposition that I don't feel precisely captures causal inference..."** You are correct that
CATE is not identifiable under unmeasured confounding and that the reason we consider CATE (and not ITE) is that
ITE is not identified without parametric assumptions. We will integrate this feedback and be more precise.

**R1 "I couldn't really follow the section on negative sampling [...] does it produce better calibrated estimates of**
**the variance of CATE?"** Negative sampling is a method specifically for CEVAE, not for the other neural methods. Its
effect is to increase the estimated uncertainty more sharply even when a point $x$ is only slightly outside the region with
good data coverage. We will clarify how this works. Epistemic uncertainty is plotted in black below:

(a) BCEVAE with negative sampling        (b) BCEVAE without negative sampling

**R1 "Eqn 4 reviews one notion of uncertainty for regression tasks. [What is its role?]"** Eq. 4 is a typical measure
for epistemic uncertainty in non-causal tasks. It is simply there to build basic background knowledge about epistemic
uncertainty, which we later extend to the task at hand. The particular decomposition of the variance in eq. 4 is just the
standard implementation. We agree that this can be clarified.

**R2 "The paper makes the assumption of ignorability [...] This is unlikely to hold in practice"** We agree with
the reviewer that the assumption is unlikely to hold in practice and that our extension to Causal Effect Variational
Autoencoders replaces this with a slightly weaker assumption. Addressing true hidden confounding is beyond the scope
of this paper and we leave it to future work.

**R2 "results in the paper seem to sweep over some threshold [ [...] where or how this threshold should be set?"**
First, thank you for pointing out arxiv:1903.12220, it is relevant to this question and will be added to our lit survey.
In general, setting the threshold will be a domain-specific problem that depends on the cost of type I (incorrectly
recommending treatment) and type II (incorrectly withholding treatment) errors. We would appeal to domain experts to
ascertain such costs. For example, in lung cancer screening (PMC4817217), the CT scan is a covariate, whether to
obtain follow-up scans is the treatment, and death due to lung cancer is the outcome. Here, the cost of a type II error is
much higher and would need to be accounted for in determining the threshold. In the diagnostic setting, thresholds
have been set to satisfy public health authority specifications on sensitivity and specificity; e.g. for diabetic retinopathy
detection (nature.com/articles/s41598-017-17876-z). When deployed, the treatment recommendations are given for
novel individuals; therefore, thresholds will need to be determined using the data available at training time.

**R3 Authors should add more detail to section 3** Currently we introduce the rejection policies in section 6 and give
details in the appendix. Following the reviewer's suggestion, we will move this to section 3.

**R3 "How can the epistemic uncertainty estimation formula (7) be applied to CEVAE, for which both w and z**
**are random?"** Great point, this should be clarified. Instead of just sampling the parameters w0,w1, we sample these
parameters *and* z, independently. z must be sampled too as this is what the standard CEVAE does (see eq. 8).

**R3 "the predictive uncertainty policy and the epistemic uncertainty policy appear to behave very similarly?**
**Are there data / model scenario that we would prefer one over the other?"** The predictive policy is shown for
comparison; it is not part of our method. The policies behave the same when there is no aleatoric/label noise (like with
MNIST) because then aleatoric uncertainty is zero. But when we do have aleatoric/label noise (like with IHDP) they
can behave differently (predictive being worse). Whenever the task is to estimate CATE, we must use the epistemic
uncertainty because the uncertainty in CATE is only epistemic (second r.h.s. term of eq. 7). The aleatoric uncertainty
component would only matter for estimating the (unidentifiable) individual treatment effect (ITE).

**R4 "the core value behind this work is an effort to handle no-overlap [...] Yet, it's a bit hard to say their proposed**
**methodology is quite novel in terms of technicality."** We agree that part of the contribution is an effort to handle the
no-overlap situation and covariate shift by using the combination of modeling uncertainty and causal effect estimation,
both in VAEs and with a plethora of other SOTA approaches in the causality field. To the best of our knowledge, we
are the first to bring these methods together and the first to empirically show their value for this task. We believe that
the fact that we arrive at these results by integrating existing methods does not take away from the significance or the
novelty of our solution, as noted by R1, R2 and R3.

[Meta-Review · NeurIPS 2020]

The reviewers unanimously recommend the paper to be accepted. While the paper was well understood, please consider their suggestions for improving its clarity. In this regard, a hyphen (causal-effect inference instead of causal effect inference) would help better understand the title, since in a machine learning conference, causal most often refers to inference.